# ADAPTIVE LOG-EXP PERTURBATIONS FOR SECURE AI IMAGE COMPRESSION

## ABSTRACT

AI image compression has outperformed traditional methods in both efficiency and quality but remains vulnerable to adversarial attacks. Most attacks on deep neural networks (DNNs) involve adding small perturbations to the input image to deceive the system and produce incorrect results. While simple, these additive perturbations affect pixels uniformly across different intensity levels, from dark to bright regions. However the human eye is less sensitive to variations in dark areas than in bright ones, making noise in brighter areas more visible. This observation suggests a novel attack strategy that minimizes the visibility of adversarial noise through adaptive perturbations. To achieve this, we propose a nonlinear log-exp perturbation, which applies more noise to dark pixels while minimizing its impact on bright areas.

We evaluated this perturbation model in two scenarios: one distorts the output of decompression models and another one increases the bit rate of compressed images without visibly affecting quality. Our findings offer new strategies to protect AI-driven image compression systems, ensuring both security and performance in practical applications.

## 1 INTRODUCTION

The advent of Deep Neural Networks (DNNs) and Variational Autoencoders (VAEs) Kingma & Welling (2014) has brought significant improvements into image compression. These technologies enable the encoding of images into a compact latent space, facilitating efficient storage and transmission Ballé et al. (2016). However, these AI-based models have their own challenges. Deep neural networks (DNNs) are generally large. For example, the Cheng2020-anchor model has a size of 120 MB, while the Attention TCM for AI compression has a size of almost 900 MB. Larger models are more vulnerable to adversarial attacks. This vulnerability is demonstrated in the work by Tong Chen and Zhan Ma Chen & Ma (2021), which shows that AI compression models can be attacked easily with simple additive perturbations and the Projected Gradient Descent (PGD) method.

Adversarial attacks have become an important area of research in the field of deep learning. The vulnerability of these models to specifically crafted perturbations is well-known.

For attacking AI compression models, one can adapt many existing methods for adversarial attacks from other domains of applications, such as classification, text, and music. However, these methods do not account for the characteristics of the human eye.

Most current adversarial attack noise models utilize simple additive noise, often without considering the human visual system's perception. This results in perturbations that may be overly visible to the human eye, undermining the stealthiness of the attack. However, the human eye does not perceive noise uniformly across different luminance levels. Accounting for this can lead to more perceptually imperceptible yet effective adversarial attacks.

In order for the adversarial attacks to succeed, the need to craft perturbations which are invisible to human eyes is important. In this paper, inspired by Weber's Law for light incremental threshold, we propose a new nonlinear perturbation model which is based on the log-exp function, can adapt to the luminance of different regions in an image. This approach allows for more imperceptible adversarial attacks, as the noise generated follows the light incremental threshold of the human visual system.

Our method demonstrates that by aligning noise generation with the properties of human perception, adversarial attacks can be made less detectable without compromising their effectiveness. This opens new avenues for the development of advanced adversarial techniques that take human visual perception into account, ensuring that perturbations are optimized for both efficacy against models and invisibility to humans.

## 2 Neural Image Compression Framework

The application of deep learning to image compression has significantly advanced the field, particularly through the use of autoencoder-based architectures. Balle et al. Ballé et al. (2018) proposed an end-to-end variational autoencoder (VAE) model, compressing image representations into a latent distribution and using a hyperprior to capture spatial dependencies. The framework optimizes rate-distortion performance by balancing bit rate (entropy) and image quality (distortion), often using Mean Squared Error (MSE) or Multi-Scale Structural Similarity (MS-SSIM) as metrics. Minnen et al. Minnen et al. (2018) extended this with an autoregressive context model, while Cheng et al. Cheng et al. (2020) further improved performance using Gaussian Mixture Models for more precise latent representation estimation. Specifically,they introduced two enhanced architectures: the "Cheng-anchor2020" model, which incorporates residual blocks in the analysis and synthesis transforms, and the "Cheng-Atten" model, which combines both residual and attention modules in these transforms.

In Liu et al. (2023), the Transformer-CNN Mixture (TCM) model combines CNNs' local feature modeling with Transformers' non-local capabilities, achieving state-of-the-art rate-distortion performance through an efficient hybrid design with Swin-transformer-based attention modules.

In these frameworks, the compression process involves transforming the image into a latent space, quantizing the latent variables, and reconstructing the image from the quantized representation. The optimization objective minimizes both the rate (bit usage) and distortion (image quality loss), making these methods highly effective for compressing images without noticeable visual degradation.

### 2.1 Adversarial Attack Methods

Adversarial attacks, particularly those utilizing additive perturbations, pose a serious threat to neural image compression systems by introducing carefully crafted noise that can degrade the model's performance. These attacks aim to subtly alter the input image to either reduce compression efficiency or impair the reconstructed output quality, often without perceptible changes to the human eye. The most common attack methods include: Common attack methods include FGSM Goodfellow et al. (2014), which adds noise in the gradient direction, PGD Madry et al. (2019) for iterative refinement, BIM Kurakin et al. (2018) for repeated perturbations, Carlini and Wagner Lin et al. (2021) for optimization-based minimal perturbations, and Wasserstein Attack Wu et al. (2020) for semantically meaningful perturbations.

These methods are widely used due to their simplicity and effectiveness in generating adversarial examples. However, they face several challenges: perturbations may be more visible in bright regions, uniform noise application can lead to suboptimal attacks, and large perturbations can noticeably degrade image quality. Additionally, these attacks are often vulnerable to defense techniques such as adversarial training or preprocessing.

For neural image compression systems, these challenges are particularly critical, as visible perturbations can disrupt the compression process and compromise image quality in visually sensitive applications. As a result, this motivates us to develop advanced attack strategies that consider regional sensitivity and compression-specific characteristics to improve the stealth and effectiveness of adversarial attacks in this domain.

### 2.2 Robustness of AI Image Compression

In Liu et al. (2022), the authors explored the robustness of deep learning-based image compression models under adversarial attacks. They applied both white-box and black-box attacks to increase the bitrate of compressed images significantly. Using a white-box approach with FGSM, they achieved up to a 50% bitrate increase but applied perturbations globally across the entire image, resulting in

highly visible artifacts. In the black-box setting, their DCT-Net achieved a 4x increase in bitrate at best, but again, the perturbations were clearly noticeable.

However, their work has notable limitations. No perceptual similarity metrics, such as PSNR or SSIM, were used to evaluate how closely the attacked images resembled the originals. This is a critical gap, as such metrics would provide a clearer picture of attack impact beyond bitrate changes. Additionally, their approach does not address localized attacks, which could potentially lead to more imperceptible perturbations with similar effectiveness.

Lei et al. (2021) explored out-of-distribution OOD-robust compression, using distributionally robust optimization and structured coding to handle distribution shifts. However, it did not address adversarial attacks or the challenge of ensuring imperceptible perturbations, focusing solely on OOD scenarios.

A recent study introduced benchmarks (CLIC-C and Kodak-C) and spectral inspection tools to evaluate the out-of-distribution (OOD) robustness of neural image compression (NIC) models Lieberman et al. (2023), revealing key insights into their performance under distribution shifts. The work highlighted NIC's ability to handle high-frequency corruptions better than classic codecs but noted challenges in generalizing to high-frequency shifts. Unlike our focus, this study did not explore adversarial attacks designed to induce artifacts or increase bpp in NIC models.

## 3  NONLINEAR PERTURBATION

### 3.1  NOISE MODELING AND HUMAN VISUAL SENSITIVITY

Most current adversarial attack noise models utilize simple additive noise, often without considering the human visual system's perception. This results in perturbations that may be overly visible to the human eye, undermining the stealthiness of the attack. However, the human eye does not perceive noise uniformly across different luminance levels. Accounting for this can lead to more perceptually imperceptible yet effective adversarial attacks.

A key principle that describes the sensitivity of the human visual system is Weber's Law Weber (1834). According to this law, the just noticeable difference (JND) in stimulus intensity, or the light incremental threshold ($\delta I$), is proportional to the background intensity ($I$). Specifically, for low luminance levels (darker areas), the ratio $\delta I / I$ is relatively large. As luminance increases, this ratio becomes smaller and tends to remain constant for mid-range luminance levels between 1 and 100 millilamberts. Brightness discrimination is poor (large Weber ratio) at low illumination levels and improves significantly as $I$ increases.

To illustrate this, consider the following examples of luminance and their corresponding threshold ratios:

- For a luminance of 0.001 mL, the threshold ratio $\frac{\delta I}{I}$ is approximately 0.2.
- For a luminance of 0.01 mL, the threshold ratio decreases to around 0.1.
- For a luminance of 1 mL, the threshold ratio is about 0.02.

This relationship highlights that the human visual system is less sensitive to brightness changes in dark regions compared to brighter regions, where the eye struggles to discern small variations. The dashed line in Figure 238 in Rutten & van Venrooij (2024) illustrates this concept.

### 3.2  VISUAL PERCEPTION AND ADAPTIVE PERTURBATIONS

The Weber-Fechner law Fechner (1860) further describes the non-linear relationship between stimulus intensity ($I$) and perceived sensation ($S$)

$$S = 2.3k \log_{10} I + C \tag{1}$$

where $k$ determines the steepness of the curve and $C$ defines its vertical position. This law explains why the eye's sensitivity to brightness changes varies across luminance levels. In very low luminance environments, background noise in the eye makes it difficult to detect small changes in light, while in

extremely bright conditions, the eye becomes overwhelmed and loses sensitivity to minor differences in luminanceRutten & van Venrooij (2024) .

This insight suggests that by accounting for the varying sensitivity of the human eye, adversarial noise can be better tailored to different regions of an image.

### 3.3 Log-Exp Noise Model Inspired by Weber's Law

Building upon this understanding of human visual sensitivity, we propose a novel adversarial noise model that leverages the light incremental threshold. The core idea is to generate noise that adapts to the luminance of different regions within an image, ensuring that perturbations are less detectable by the human eye while still effective in deceiving neural networks.

Specifically, we propose to model the adversarial noise as a function of the luminance $I$ and the random noise generated for a given pixel as follows

$$I' = \log(\exp^I + n), \tag{2}$$

where $n$ is small perturbation noise, and $I'$ is the perturbed intensity from the luminance level $I$ of the pixel. Note that the noise $n$ is usually small, ensuring that the logarithm function does not encounter errors.

The Taylor expansion of the log-exp function in (2) around $n = 0$

$$I' = \log(\exp^I + n) = I + n \exp(-I) + O(n) \tag{3}$$

shows that $\delta I = n \exp(-I)$ represents the additive perturbation added to the pixel value $I$ (luminance level). The perturbation $\delta I$ adaptively changes to the pixel values $I$ as an exponential decay factor $\exp(-I)$. Note that the normalized pixel value $I$ is in the interval of $[0, 1]$. In darker regions, $I \approx 0$, the perturbation $\delta I$ is as the noise $n$, but it monotonically decreases as $I$ increases to 1, yielding $\delta I \approx 0.3679 \, n$, i.e., introducing less noise to the bright region.

This exponential decay model ensures that in darker regions, where the human eye is less sensitive to small perturbations, the noise can be slightly stronger. Conversely, in brighter regions, the noise is minimized to remain imperceptible.

The perturbation $p(I)$ can be modified with a decaying factor $\kappa$

$$\delta I = n \cdot \exp(-\kappa I) \tag{4}$$

where $\kappa$ can range from 1 to 3. This function grows quickly for large values of $I$ (bright areas) and slows for smaller values of $I$ (dark areas), allowing for more control over perturbations in different luminance regions.

### 3.4 Just Noticeable Difference Models

Several existing models have explored the concept of JND to minimize the visibility of noise in images. For instance, Hu et al. (2023) incorporate color sensitivity to adjust the sub-JND thresholds of Y, Cb, and Cr components, creating a color-sensitivity-based JND model (CSJND). This model reflects the visibility limitations of the human visual system and is commonly applied to perceptual image and video processing.

Another example is the Just Noticeable Difference Model Zhang et al. (2023), which focuses on estimating the JND based on the characteristics of the human visual system. This model considers spatial contrast sensitivity functions and other factors to create a more accurate representation of perceptual thresholds. These models aim to create noise that minimizes visibility, aligning with the main idea of our proposed method.

In the context of adversarial attacks, the goal is to introduce small perturbations to an image that cause a neural network to misclassify the image or degrade its performance, but without making the perturbations visible to humans. Our adaptive noise model, guided by Weber's law, is designed to achieve this by ensuring that noise is proportional to the sensitivity of the human eye to brightness changes in different regions of the image.

By incorporating principles from Weber's law, specifically the relationship between luminance and detection thresholds, we introduce a more sophisticated noise model that adapts to the luminance of

different regions in an image. This approach allows for more imperceptible adversarial attacks, as the noise generated follows the light incremental threshold of the human visual system.

## 4 MAXIMIZING DISTORTION IN ADVERSARIAL ATTACKS ON IMAGE COMPRESSION

Following on the log-exp perturbation model proposed in the previous section, we propose a max-distortion adversarial attack aimed at finding a noise pattern that maximizes the distortion in decompressed images.

Consider an original image $\boldsymbol{x}$, and let $\boldsymbol{x}^\star$ be the perturbed version, defined as $\boldsymbol{x}^\star = \log(\exp(\boldsymbol{x}) + \boldsymbol{n}_a)$, where $\boldsymbol{n}_a$ represents the adversarial noise for the attack. The neural network's decompressed output is denoted by $\hat{\boldsymbol{x}} = f(\boldsymbol{x}^\star)$. Similar to adversarial examples used in classification, the goal here is to learn a noise pattern $\boldsymbol{n}_a$ that minimally alters $\boldsymbol{x}$ but significantly impairs the compression model's output quality. This is done by amplifying the difference between the decompressed images $f(\boldsymbol{x}^\star)$ and $f(\boldsymbol{x})$ or between $f(\boldsymbol{x}^\star)$ and $\boldsymbol{x}$ .

The attack objective can be formulated as

$$\min_{\boldsymbol{n}_a} \quad PSNR(f(\boldsymbol{x}^\star), \boldsymbol{x}) + \lambda \|\boldsymbol{n}_a\|_1 \quad \text{s.t.} \quad \|\boldsymbol{n}_a\|_\infty \leq \delta, \tag{5}$$

where $\|\boldsymbol{n}_a\|_\infty$ represents the infinity norm, and $\delta > 0$ defines the allowable noise level. The noise pattern $\boldsymbol{n}_a$ is reparameterized as $\boldsymbol{n}_a = \delta \tanh(\kappa \cdot \boldsymbol{u})$, where $\boldsymbol{u}$ is an unconstrained variable and $\kappa$ controls the sharpness of the noise transition, ensuring that $\boldsymbol{n}_a$ approaches $\pm\delta$ without reaching those bounds.

To maximize distortion while maintaining imperceptibility, the attack focuses on local high-entropy regions, which often contain more detail and are more sensitive to perturbations. A binary mask identifies these regions by grouping similar pixels and selecting superpixels with the highest entropy, indicating complex areas of the image. The mask marks significant regions for targeted distortion.

The optimization problem in (5) is rewritten as

$$\min_{\boldsymbol{u}} \quad PSNR(f(\boldsymbol{x}^\star), \boldsymbol{x}) + \lambda \|\boldsymbol{u}\|_1, \tag{6}$$

where $\boldsymbol{x}^\star = \log(\exp(\boldsymbol{x}) + \delta \cdot \texttt{mask} \cdot \tanh(\boldsymbol{u}))$, $\texttt{mask}$ represents the binary mask of the targeted attack region.

To solve this optimization, methods such as Stochastic Gradient Descent (SGD) or ADAM are used to estimate $\boldsymbol{u}$. Larger values of $\boldsymbol{u}$ push the noise pattern $\boldsymbol{n}$ near the boundary, while reducing large coefficients helps avoid local minima during optimization.

The proposed approach concentrates the attack on high-entropy regions, where image content is more unpredictable. By constraining the noise using a nonlinear transformation, the perturbations are significant enough to degrade image quality but remain imperceptible to the human eye, balancing effectiveness and subtlety.

The mask is smoothed using a Gaussian filter with $\sigma = 21$, which helps regulate the noise and progressively reduce it toward the desired threshold. Through iterative updates, the mask is gradually shrunk, focusing the attack on a smaller region and refining the perturbations for maximal distortion with minimal visibility. For more detailes how the hyperparameters were selected please refer to A

As an example for the proposed method, Figure 1 demonstrates the noise filtering process for attacking the image kodim19, compressed using the Cheng2020-anchor model. It shows the final attacked image with the shrunken mask applied, the decompressed attacked image where artifacts are visible, the corresponding noise mask, and the step-by-step progression leading to the final shrunken mask.

## 5 DEFENSE STRATEGIES

The goal of the defense strategy is to reduce the distortion $D$ (maximizing PSNR) between the attacked image $\boldsymbol{x}^\star$, which has been compromised by adversarial perturbations, and the decompressed output image $\boldsymbol{x}_{out} = f(\boldsymbol{x}^\star)$, which may contain significant artifacts. Without prior knowledge of

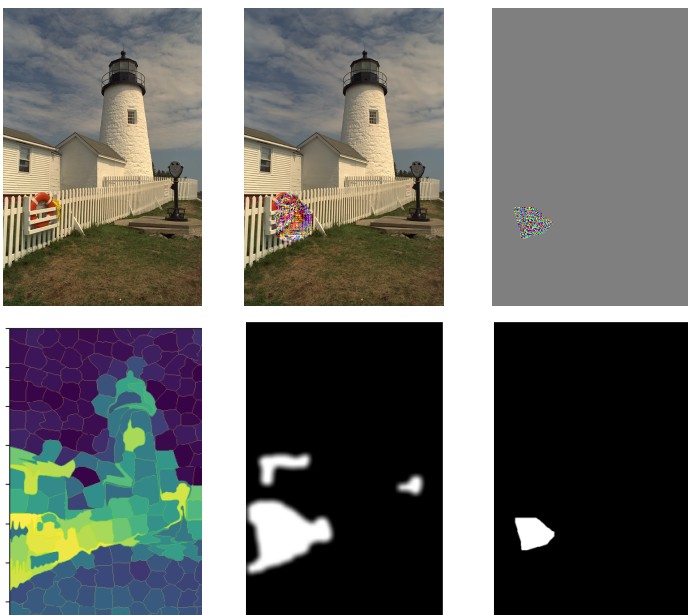

Figure 1: kodim19, Cheng2020-anchor, Maxdistortion attack. Top: the attacked image, model's output (decompressed image), attack noise pattern. Bottom: entropy regions, smoothed mask, final mask. **PSNR(oi, oo) = 36.85 dB**, **PSNR(ai, ao) = 20.62 dB** , **BPP(oc) = BPP(ac) = 0.85**, where oi, oo, oc - original (input, output, compressed) image, and ai, ao, ac - attacked (input, output, compressed) image.

the attack model, the defense aims to add corrective noise to the attacked image rather than recovering the original perturbation. Specifically, we focus on learning an optimal noise pattern for defense, $\boldsymbol{n}_d$, such that when added to the attacked image, it enhances the quality of the decompressed output.

The defense problem can be formulated as the following optimization task

$$\min_{\mathbf{n}_d} \quad D\left(\boldsymbol{x}^\star, f(\boldsymbol{x}^\star + \boldsymbol{n}_d)\right) + \lambda\|\boldsymbol{n}_d\|_1, \quad \text{s.t.} \quad \|\boldsymbol{n}_d\|_\infty \leq \delta, \tag{7}$$

or

$$\max_{\mathbf{n}_d} \quad PSNR(\mathbf{x}^\star, f(\mathbf{x}^\star + \boldsymbol{n}_d)) + \lambda\|\boldsymbol{n}_d\|_1 \quad \text{s.t.} \quad \|\boldsymbol{n}_d\|_\infty \leq \delta, \tag{8}$$

where $f(\boldsymbol{x}^\star + \boldsymbol{n}_d)$ represents the decompressed image after adding the corrective noise $\boldsymbol{n}_d$. The parameter $\delta$ defines the upper bound for the magnitude of the noise, while $\lambda$ controls the regularization for the sparsity constraints on the corrective noise pattern $\boldsymbol{n}_d$.

As in previous approaches, the noise $\boldsymbol{n}_d$ is modeled as a hyperbolic tangent transformation of unconstrained parameters $\boldsymbol{u}$, expressed as $\boldsymbol{n}_d = \delta \tanh(\boldsymbol{u})$. This ensures that the noise stays within the predefined bounds. Stochastic Gradient Descent (SGD) is used to optimize the noise pattern, aiming to reduce artifacts and enhance image quality.

Once the optimal noise $\boldsymbol{n}_d$ is learned, it is added to the attacked image $\boldsymbol{x}^\star$, yielding a refined image $\boldsymbol{x}_{def} = \boldsymbol{x}^\star + \boldsymbol{n}_d$. Compressing this refined image through the model results in an output with significantly reduced artifacts, thereby improving the overall quality of the decompressed image.

Follow up the same example for the max-distortion attack, Table 1 presents the results of our defense strategy for kodim19, where we achieved a PSNR($\boldsymbol{x}^\star, f(\boldsymbol{x}_{def})$) of 36.75 dB, which is very close to the baseline. Additionally, the defended image closely resembles the original one, as confirmed by the results of other metrics. Figure 2 provides a visual illustration of the defense process.

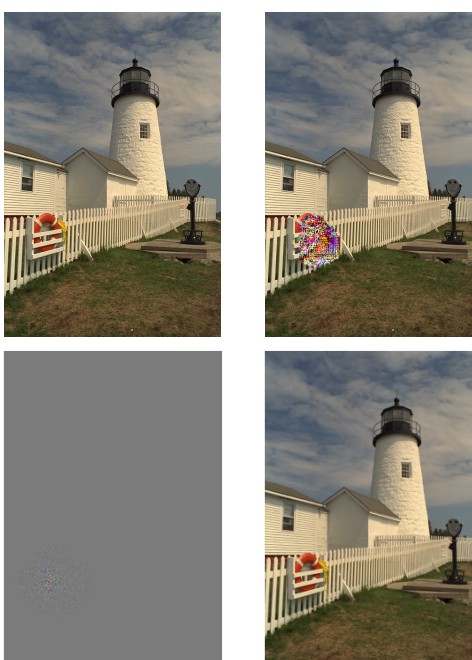

Figure 2: Sequential visualization of the defense method for the kodim19 image infected by MaxDistortion attack. Top: the attacked image and its decompressed image with disrupted pattern. Bottom: noise pattern learnt by the defense method, and the decompressed image after removing the learnt noise. The decompressed image after noise removal from the infected image has VIF = 0.9883.

## 6 RESULTS

We evaluated the proposed attack and defense algorithms on the Kodak dataset Kodak (1993), a widely recognized benchmark for image quality assessment. The AI compression models used in our experiments were sourced from the InterDigital CompressAI library Bégaint et al. (2020). All experiments were conducted within the PyTorch framework, utilizing an Ubuntu server equipped with an NVIDIA A100 GPU and 32 GB of RAM. In this section, we present the experimental results for kodim images, demonstrating the performance of our attack and defense methods using both the Cheng2020-anchor and TCM compression models.

### 6.1 ATTACKS AND DEFENSES ON CHENG2020-ANCHOR MODEL

Table 1 presents the results at each step of our proposed method. The initial significant drop in PSNR is observed in the first step, where the Log-Exp noise function is applied to the mask derived from the high-entropy filter. This filter identifies regions with more details in the image. Despite the PSNR reduction, the refinement process involving mask smoothing and shrinking leads to better results in terms of `PSNR(oi, ai)` where `ai` is the attacked image, i.e., $x^\star$, and `oi` is the original image, i.e., $x$, with a 16 dB difference in `PSNR(ai, ao)` where `ao` is the attacked output, compared to the baseline. Notably, the PSNR between the attacked and original image is higher when using the mask shrink step, indicating that the attacked image appears more realistic. This observation is further supported by the VIF metric, which confirms the visual quality preservation. In comparison, the additive noise attack results in more visible noise, with a 4 dB reduction in the PSNR of the attacked image, making it less realistic than in the Log-Exp noise case.

Figure 3 shows the comparison between the two methods of applying noise, Additive Noise and Log-Exp Noise, using the Cheng anchor model with quality 6. We observe that PSNR(oi, ai) values demonstrate that Log-Exp Noise introduces less distortion to the original image compared to Additive Noise, making the attacked image appear more realistic with less visible perturbations. In terms of PSNR(ai, ao), both methods show significant degradation after decompression. However,

Table 1: Comparison of PSNR (dB), BPP, SSIM, and VIF metrics at different stages of our log exp method vs additive noise for kodim19. Abbreviations: oi, oo, oc - original (input, output, compressed) image; ai, ao, ac - attacked (input, output, compressed) image.

| Method | PSNR(ai, ao) | PSNR(oi, ai) | BPP(ac) | SSIM(ai, oi) | VIF(oi, ai) | VIF(oi, ao) |
|---|---|---|---|---|---|---|
| baseline_full | 36.85 | | 0.85 | 0.9718 | 1.00 | 1.00 |
| minpsnr_ | 19.33 | 41.10 | 0.90 | 0.9503 | 0.98 | 0.45 |
| highentropy_minpsnr_masksmooth | 18.59 | 45.06 | 0.87 | 0.9487 | 0.91 | 0.40 |
| highentropy_minpsnr_maskshrink | 20.62 | 50.67 | 0.85 | 0.9567 | 0.98 | 0.53 |
| highentropy_minpsnr_maskshrink_additive | 27.42 | 46.67 | 0.86 | 0.9681 | 1.00 | 0.87 |
| def_minpsnr | 36.75 | 51.60 | 0.85 | 0.9717 | 1.00 | 1.00 |

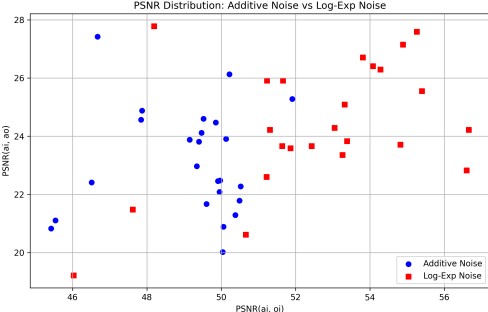

Figure 3: Comparison between Additive Noise and Log-Exp Noise using the Cheng anchor model with quality 6

although Additive Noise degrades the performance more than Log-Exp Noise, our main objective is to make the applied noise less noticeable. Therefore, the Log-Exp Noise method achieves this goal more effectively by keeping the perturbations subtle and less perceptible while maintaining a higher image quality. Table 2 summarize the results of our attack method using Cheng-anchor model. The PSNR drop between the attacked image `ai` and the decompressed attacked image `ao` varies across quality levels. At the compression quality level q1, the PSNR drop ranges from 4.28 dB to 10.98 dB, indicating a significant degradation in image quality. For the compression quality of 3, the drop is slightly less severe, ranging from 6.10 dB to 8.99 dB, while at the compression quality of 6, the drop ranges from 5.13 dB to 19.88 dB, with some images experiencing much larger reductions. Despite the notable decline in PSNR, the BPP values remain consistent between the original and attacked images, indicating that the attack does not significantly alter the file size. This demonstrates that the attack is effective in degrading image quality while maintaining the same compression characteristics.

We observe an average drop in PSNR(ai, ao) of -5.25, -3.5, and -12.21 compared to the baseline PSNR for quality levels q1, q3, and q6, respectively. Notably, the BPP remains unchanged before and after the attack, aligning with our objective to maintain the original file size while introducing minimal noise(invisible to the human eye). This targeted noise application is designed to effectively reduce PSNR, thereby introducing perceptible artifacts in the decompressed output of the attacked image.

The results summarized in Table 3 illustrate the performance of our defense strategy against the maxdistortion attack. Here, d_i refers to the defended image input fed into the AI compression model, while d_o denotes its decompressed output produced by the same model. Notably, we observed that the PSNR values of the defended images closely approach those of the original images, indicating that our defense mechanism is effective in preserving image quality. Additionally, the BPP(dc) values show that the file sizes of the defended images remain consistent with those of the original images, highlighting the efficiency of our method.

Moreover, the SSIM (Structural Similarity Index) values further reinforce the notion that the defended images retain structural similarities to the originals.

Table 2: MaxDistortion attack on Cheng2020-anchor. Comparison of PSNR(ai, ao) - BPP(ac) pairs across three different quality levels: q1, q3, and q6, which represent increasing quality levels, respectively (ai, ac, ao - attacked: input, compressed, output images).

| Image name | original (q1) | | attacked (q1) | | original (q3) | | attacked (q3) | | original (q6) | | attacked (q6) | |
|---|---|---|---|---|---|---|---|---|---|---|---|---|
| | PSNR | BPP | PSNR | BPP | PSNR | BPP | PSNR | BPP | PSNR | BPP | PSNR | BPP |
| kodim01 | 26.29 | 0.25 | 23.79 | 0.26 | 29.19 | 0.53 | 24.62 | 0.53 | 35.22 | 1.42 | 25.92 | 1.42 |
| kodim02 | 30.37 | 0.14 | 25.37 | 0.14 | 32.32 | 0.22 | 25.43 | 0.23 | 36.97 | 0.69 | 23.66 | 0.70 |
| kodim03 | 31.91 | 0.13 | 25.04 | 0.13 | 34.58 | 0.21 | 25.99 | 0.21 | 39.49 | 0.51 | 24.22 | 0.51 |
| kodim04 | 30.19 | 0.15 | 25.33 | 0.15 | 32.61 | 0.25 | 25.89 | 0.25 | 37.28 | 0.70 | 27.78 | 0.71 |
| kodim05 | 26.65 | 0.32 | 22.91 | 0.32 | 29.81 | 0.57 | 25.18 | 0.57 | 35.77 | 1.34 | 26.41 | 1.34 |
| kodim06 | 27.71 | 0.22 | 22.51 | 0.23 | 30.54 | 0.40 | 56.96 | 0.40 | 36.58 | 1.07 | 26.29 | 1.07 |
| kodim07 | 31.10 | 0.18 | 23.44 | 0.19 | 34.22 | 0.27 | 26.22 | 0.28 | 39.29 | 0.60 | 23.36 | 0.61 |
| kodim08 | 26.43 | 0.35 | 23.32 | 0.35 | 29.14 | 0.58 | 25.21 | 0.57 | 34.81 | 1.44 | 24.29 | 1.44 |
| kodim09 | 31.58 | 0.15 | 25.39 | 0.15 | 34.36 | 0.22 | 26.58 | 0.24 | 38.69 | 0.52 | 25.09 | 0.52 |
| kodim10 | 31.35 | 0.16 | 26.11 | 0.17 | 34.16 | 0.24 | 26.44 | 0.25 | 38.58 | 0.56 | 24.22 | 0.56 |
| kodim11 | 28.58 | 0.19 | 22.21 | 0.20 | 31.18 | 0.34 | 27.97 | 0.35 | 36.60 | 0.93 | 19.22 | 0.94 |
| kodim12 | 31.52 | 0.13 | 24.56 | 0.14 | 33.78 | 0.20 | 26.77 | 0.21 | 38.46 | 0.57 | 22.83 | 0.57 |
| kodim13 | 24.36 | 0.36 | 21.83 | 0.36 | 26.70 | 0.69 | 20.96 | 0.70 | 32.40 | 1.82 | 25.91 | 1.82 |
| kodim14 | 27.39 | 0.23 | 23.23 | 0.24 | 30.20 | 0.43 | 26.45 | 0.44 | 35.60 | 1.17 | 23.66 | 1.17 |
| kodim15 | 30.44 | 0.15 | 26.56 | 0.15 | 32.66 | 0.24 | 26.64 | 0.26 | 37.41 | 0.65 | 22.60 | 0.65 |
| kodim16 | 29.82 | 0.15 | 23.53 | 0.16 | 32.45 | 0.26 | 26.01 | 0.27 | 38.02 | 0.75 | 23.75 | 0.75 |
| kodim17 | 30.43 | 0.16 | 23.88 | 0.17 | 33.01 | 0.26 | 25.82 | 0.27 | 37.77 | 0.64 | 25.55 | 0.64 |
| kodim18 | 26.87 | 0.24 | 23.76 | 0.25 | 29.55 | 0.44 | 24.62 | 0.46 | 34.43 | 1.15 | 27.59 | 1.15 |
| kodim19 | 29.28 | 0.18 | 21.68 | 0.18 | 31.62 | 0.30 | 25.71 | 0.30 | 36.85 | 0.85 | 20.62 | 0.85 |
| kodim20 | 31.17 | 0.15 | 23.66 | 0.16 | 33.46 | 0.22 | 26.11 | 0.23 | 38.36 | 0.59 | 26.71 | 0.59 |
| kodim21 | 28.25 | 0.21 | 22.52 | 0.22 | 31.13 | 0.37 | 26.51 | 0.38 | 36.56 | 0.92 | 27.15 | 0.92 |
| kodim22 | 28.43 | 0.17 | 22.34 | 0.18 | 31.01 | 0.32 | 25.87 | 0.33 | 36.09 | 0.94 | 21.48 | 0.95 |
| kodim23 | 32.66 | 0.14 | 22.13 | 0.16 | 35.16 | 0.19 | 24.51 | 0.20 | 39.35 | 0.43 | 23.59 | 0.44 |
| kodim24 | 26.98 | 0.26 | 23.73 | 0.26 | 29.46 | 0.45 | 25.10 | 0.46 | 35.16 | 1.13 | 23.83 | 1.13 |
| **Average** | **29.25** | **0.20** | **24.00** | **0.21** | **31.60** | **0.36** | **28.09** | **0.37** | **36.71** | **0.93** | **24.50** | **0.94** |

By comparing the average results of the defense from Table 3 with the average results of the original from Table 2, we observe that the PSNR for the defended image (PSNR(di, do) = 36.81) is very close to the baseline PSNR of the original image (PSNR = 36.71 for q6). Additionally, the high PSNR value (PSNR(di, oi)) further ensures the similarity between the defended and original images. We also note that the BPP for the defended image (0.90) is nearly identical to that of the original image (0.93).

## 6.2 ATTACKS AND DEFENSES ON TCM MODEL

We aimed to demonstrate that our adversarial attack method is effective against the latest state-of-the-art AI compression models. As a representative example, we selected the TCM model, with the highest quality variant, the N128 architecture, Liu et al. (2023).

We selected Kodak images 2, 5, and 23 for their diverse characteristics: image 2 for low-detail areas, image 5 for complex human features, and image 23 for high-detail, vibrant textures, to thoroughly test our attack and defense algorithms. By including these diverse images, we ensure that our evaluation captures a wide range of scenarios, from low to high complexity, and reflects real-world applicability for image compression and security challenges.

The average values in Table 4 demonstrate the impact of our attack and defense mechanisms. Specifically, we observe a PSNR drop from 38.66 dB (PSNR(oi, oo)) to 22.89 dB (PSNR(ai, ao)) due to the attack, representing a significant reduction of 13.77 dB. This drop is accompanied by a slight increase in bit rate, with BPP rising from 0.54 (BPP(oc)) to 0.85 (BPP(ac)). Despite this, the high PSNR(oi, ai) value of 47.21 dB indicates that the applied noise is imperceptible, maintaining a strong similarity between the original and attacked images.

Using our defense algorithm, we successfully restored image quality, achieving a PSNR(di, do) of 37.89 dB, which is very close to the original PSNR(oi, oo). Additionally, the BPP(dc) of 0.56 is nearly identical to the original BPP(oc), demonstrating that our defense algorithm effectively

Table 3: MinPSNR defenses applied to attacked images using the Cheng2020-anchor compression model at quality level 6. di, do, dc - are defended images: input, output, compressed. By "output" we refer to the decompressed image. PSNR(di, do) is the result after defense which is very close to the baseline result.

| Image | PSNR(di, do) | PSNR(di,oi) | BPP(dc) | SSIM(di,oi) | VIF(di,oi) | VIF(do,oi) |
|-------|-------------|-------------|---------|-------------|------------|------------|
| kodim01 | 35.14 | 52.16 | 1.42 | 0.9788 | 0.9990 | 0.9973 |
| kodim02 | 36.88 | 51.79 | 0.70 | 0.9610 | 0.9951 | 0.9901 |
| kodim03 | 39.46 | 57.27 | 0.51 | 0.9799 | 0.9990 | 1.0005 |
| kodim04 | 37.04 | 48.36 | 0.71 | 0.9671 | 0.9927 | 0.9928 |
| kodim05 | 35.72 | 54.53 | 1.34 | 0.9839 | 0.9995 | 0.9998 |
| kodim06 | 36.55 | 54.99 | 1.06 | 0.9778 | 0.9988 | 0.9989 |
| kodim07 | 39.22 | 54.43 | 0.61 | 0.9854 | 0.9988 | 1.0012 |
| kodim08 | 34.78 | 54.02 | 1.44 | 0.9789 | 0.9997 | 1.0014 |
| kodim09 | 38.58 | 54.16 | 0.52 | 0.9723 | 0.9982 | 0.9962 |
| kodim10 | 38.42 | 51.67 | 0.56 | 0.9729 | 0.9967 | 0.9998 |
| kodim11 | 36.34 | 47.32 | 0.94 | 0.9723 | 0.9945 | 0.9944 |
| kodim12 | 38.45 | 57.12 | 0.57 | 0.9705 | 0.9989 | 1.0005 |
| kodim13 | 32.28 | 51.46 | 1.82 | 0.9758 | 0.9991 | 0.9981 |
| kodim14 | 35.55 | 53.01 | 1.17 | 0.9737 | 0.9988 | 0.9965 |
| kodim15 | 37.29 | 52.12 | 0.65 | 0.9694 | 0.9977 | 0.9964 |
| kodim16 | 37.97 | 55.71 | 0.75 | 0.9766 | 0.9984 | 1.0010 |
| kodim17 | 37.66 | 56.56 | 0.64 | 0.9743 | 0.9989 | 1.0014 |
| kodim18 | 34.39 | 55.84 | 1.15 | 0.9674 | 0.9994 | 1.0027 |
| kodim19 | 36.75 | 51.59 | 0.85 | 0.9717 | 0.9984 | 0.9952 |
| kodim20 | 38.28 | 54.41 | 0.59 | 0.9760 | 0.9987 | 1.0011 |
| kodim21 | 36.51 | 55.57 | 0.92 | 0.9731 | 0.9992 | 1.0019 |
| kodim22 | 35.96 | 48.51 | 0.95 | 0.9670 | 0.9939 | 1.0001 |
| kodim23 | 39.16 | 52.18 | 0.44 | 0.9762 | 0.9974 | 0.9984 |
| kodim24 | 35.07 | 53.52 | 1.14 | 0.9792 | 0.9991 | 1.0074 |
| **Average** | **36.81** | **53.26** | **0.89** | **0.97** | **0.998** | **0.999** |

mitigates the attack while preserving the compressed file size. We have included example figures in the Appendix to illustrate the attack on the TCM model (see Figures 7 8).

Table 4: TCM model, N128. Log-Exp attack (full image, no mask applied) and defense. Abbreviations: oi, oo - original input and output; ai, ao - attacked input and output; di, do - defended input and output. BPP oc, ac, dc values are given for compressed image files (original, attacked, defended).

| Image | PSNR(oi, oo) | PSNR(oi, ai) | PSNR(ai, ao) | BPP(oc) | BPP(ac) | PSNR(oi, di) | PSNR(di, do) | BPP(dc) |
|-------|-------------|-------------|-------------|---------|---------|-------------|-------------|---------|
| kodim 2 | 37.74 | 45.76 | 22.95 | 0.64 | 1.14 | 37.21 | 45.81 | 0.67 |
| kodim 5 | 38.21 | 47.88 | 23.91 | 0.64 | 0.78 | 36.83 | 47.96 | 0.67 |
| kodim 23 | 40.02 | 49.98 | 21.82 | 0.34 | 0.63 | 39.63 | 50.07 | 0.35 |
| **Average** | 38.66 | 47.21 | 22.89 | 0.54 | 0.85 | 37.89 | 47.95 | 0.56 |

# 7 CONCLUSIONS

In this paper we introduced a novel adversarial attack method to impair the image compression which is based on nonlinear log-exp perturbation. To maximize distortion we adapt the Hyperbolic Transformation Method and the local high-entropy selected attack. In our experiments we have demonstrated that these techniques can effectively disrupt the compression models by significantly impacting the image quality and file size.

Our defense strategy proves to be capable of removing adversarial noise, allowing for high-quality image compression. These methods not only enhance our understanding of adversarial tactics but also introduces new applications for attack strategies to limit image compression.

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

## A  HYPERPARAMETER SELECTION

The following hyperparameters are chosen to balance attack effectiveness and imperceptibility:

- $\lambda = 0.0001$: Selected empirically to achieve an optimal trade-off between distortion minimization and perceptual similarity.
- $\sigma = 21$: The Gaussian filter's standard deviation smooths the binary mask, ensuring that the noise is distributed evenly across high-entropy regions. Lower $\sigma$ values lead to visual artifacts, while higher $\sigma$ excessively smooths the noise.
- $\kappa = 1$: This factor controls the sharpness of transitions in the reparameterized noise. While $\kappa = 1$ provides a smooth transition, sharper settings ($\kappa = 2, 3$) are also valid and may be tuned for specific use cases.
- $\delta = 0.08$: This represents the initial allowable noise level. During optimization, the noise is progressively reduced, and in most cases, it converges to the minimum value of $0.02$, which corresponds to the perceptual limit of the applied noise.

## B  MINIMAL DETECTABLE CHANGE OVER VISUAL RANGE

According to Weber's law, the ratio $\frac{\delta I}{I}$ tends to remain constant for mid-range luminance levels. Specifically, for luminance values typically between 1 and 100 millilamberts, the ratio of the Just Noticeable Difference (JND) to the original luminance remains relatively constant, as shown in Figure 4. Rutten & van Venrooij (2024)

However, for very low or very high luminance values, the JND generally follows a logarithmic relationship with the luminance, which is better modeled by the Fechner's law Fechner (1860)

## C  APPENDIX: ADDITIONAL RESULTS

We provide more experiment results in Tables 5-8 for the Cheng2020-attention, which is known more efficient than some other neural compression models. Figure 6 demonstrates the noise filtering process for attacking the image kodim23, compressed using the Cheng2020-attention model with quality 6.

Table 5: Comparison of PSNR (dB), BPP, SSIM, and VIF metrics at different stages of our log exp method vs additive noise for kodim01 compressed by Cheng2020-attention model. Abbreviations: oi, oo, oc - original (input, output, compressed) image; ai, ao, ac - attacked (input, output, compressed) image.

| Method | PSNR(ai, ao) | PSNR(oi, ai) | BPP(ac) | SSIM(ao) | VIF(oi, ai) | VIF(oi, ao) |
|---|---|---|---|---|---|---|
| baseline | 35.08 | | 1.4114 | 0.0213 | 0.9999 | 0.9974 |
| masksmooth | 29.92 | 45.20 | 1.4258 | 0.0242 | 0.9949 | 0.9531 |
| maskshrink | 29.96 | 45.20 | 1.4246 | 0.0241 | 0.9949 | 0.9549 |
| defense | 34.67 | 45.52 | 1.4245 | 0.0225 | 0.9953 | 0.9947 |

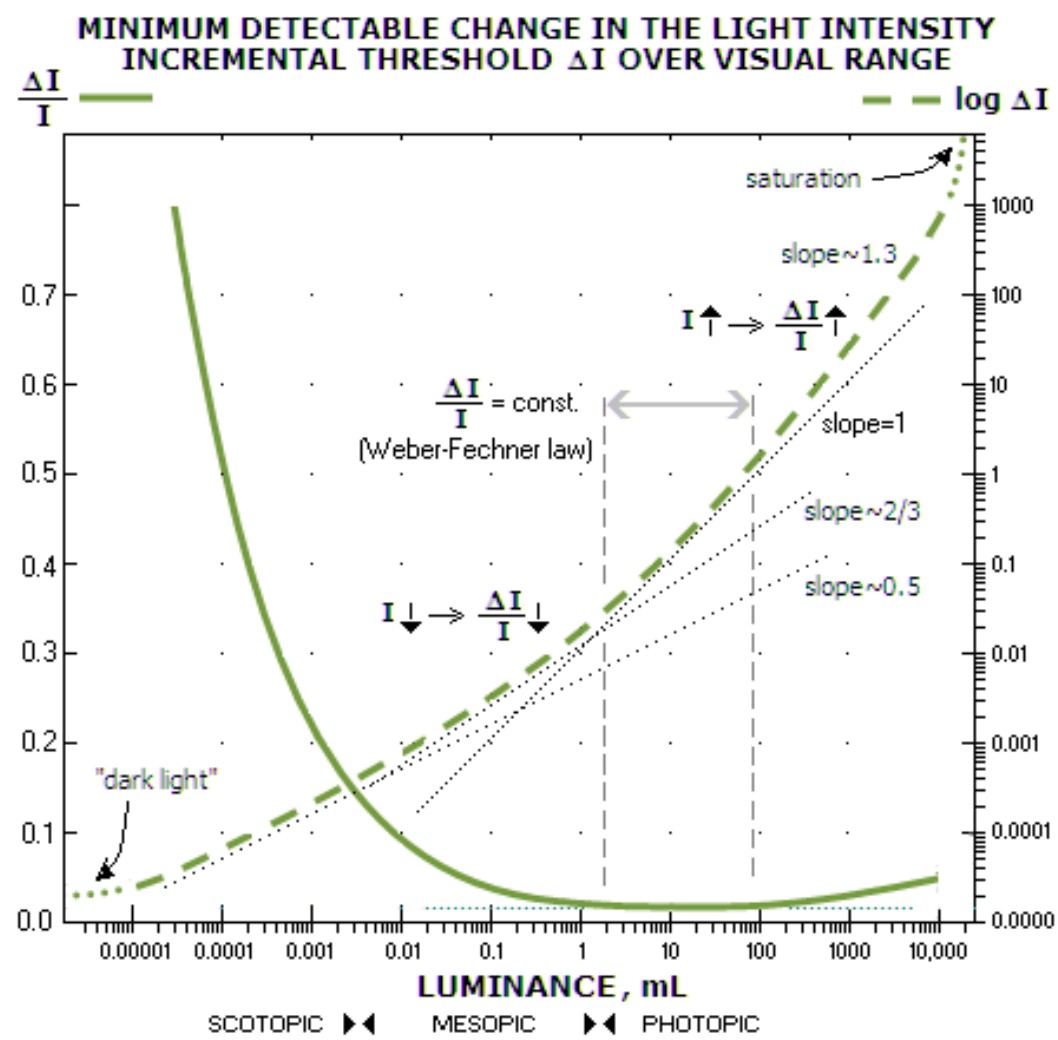

Figure 4: Minimal detectable change over visual range.

Table 6: Comparison of PSNR (dB), BPP, SSIM, and VIF metrics at different stages of our log exp method vs additive noise for kodim03 compressed by Cheng2020-attention model. Abbreviations: oi, oo, oc - original (input, output, compressed) image; ai, ao, ac - attacked (input, output, compressed) image.

| Method | PSNR(ai, ao) | PSNR(oi, ai) | BPP(ac) | M-SSIM(ai, ao) | VIF(oi, ai) | VIF(oi, ao) |
|---|---|---|---|---|---|---|
| baseline | 39.35 | | 0.5047 | 0.0204 | 0.9999 | 0.9991 |
| minpsnr | 17.79 | 36.25 | 0.6549 | 0.0594 | 0.8844 | 0.2271 |
| masksmooth | 23.09 | 43.05 | 0.5462 | 0.0318 | 0.9747 | 0.5186 |
| maskshrink | 22.78 | 46.84 | 0.5261 | 0.0297 | 0.9897 | 0.5088 |
| def_minpsnr | 38.82 | 47.49 | 0.5204 | 0.0220 | 0.9909 | 0.9950 |

Table 7: Comparison of PSNR (dB), BPP, SSIM, and VIF metrics at different stages of our log exp method vs additive noise for kodim04 compressed by Cheng2020-attention model. Abbreviations: oi, oo, oc - original (input, output, compressed) image; ai, ao, ac - attacked (input, output, compressed) image.

| Method | PSNR(ai, ao) | PSNR(oi, ai) | BPP(ac) | M-SSIM(ai, ao) | VIF(oi, ai) | VIF(oi, ao) |
|---|---|---|---|---|---|---|
| baseline | 37.25 | | 0.70 | 0.0324 | 0.9999 | 0.9967 |
| minpsnr | 21.06 | 41.76 | 0.73 | 0.0474 | 0.9686 | 0.3665 |
| masksmooth | 19.44 | 41.02 | 0.75 | 0.0517 | 0.9682 | 0.3503 |
| maskshrink | 23.97 | 48.16 | 0.71 | 0.0398 | 0.9937 | 0.5963 |
| defense | 37.03 | 49.43 | 0.70 | 0.0331 | 0.9952 | 0.9959 |

Table 8: Comparison of PSNR (dB), BPP, SSIM, and VIF metrics at different stages of our log exp method vs additive noise for kodim23 compressed by Cheng2020-attention model. Abbreviations: oi, oo, oc - original (input, output, compressed) image; ai, ao, ac - attacked (input, output, compressed) image.

| Method | PSNR(ai, ao) | PSNR(oi, ai) | BPP(ac) | M-SSIM(ai, ao) | VIF(oi, ai) | VIF(oi, ao) |
|---|---|---|---|---|---|---|
| baseline | 39.14 | | 0.43 | 0.0238 | 0.9999 | 0.9987 |
| minpsnr | 26.61 | 36.12 | 0.53 | 0.03886 | 0.9114 | 0.6616 |
| masksmooth | 29.01 | 40.96 | 0.48 | 0.03077 | 0.9677 | 0.7942 |
| maskshrink | 28.88 | 40.96 | 0.48 | 0.0309 | 0.9677 | 0.7918 |
| defense | 37.02 | 41.22 | 0.48 | 0.0283 | 0.9696 | 0.9735 |

Figure 5 demonstrates another example of the noise filtering process for attacking the image kodim24, compressed using the Cheng2020-anchor model with quality 6.

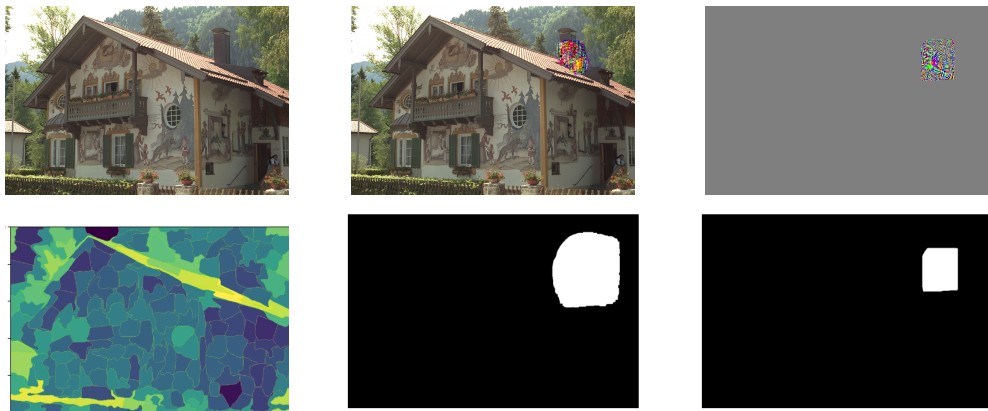

Figure 5: kodim24, Cheng2020-anchor, Maxdistortion attack. Top: the attacked image, model's output (decompressed image), attack noise pattern. Bottom: entropy regions, smoothed mask, final mask. **PSNR(oi, oo) = 35.16 dB**, **PSNR(ai, ao) = 23.83 dB**, **BPP(oc) = BPP(ac) = 1.13**, where oi, oo, oc - original input, output, compressed image, and ai, ao, ac - attacked input, output, and compressed image.

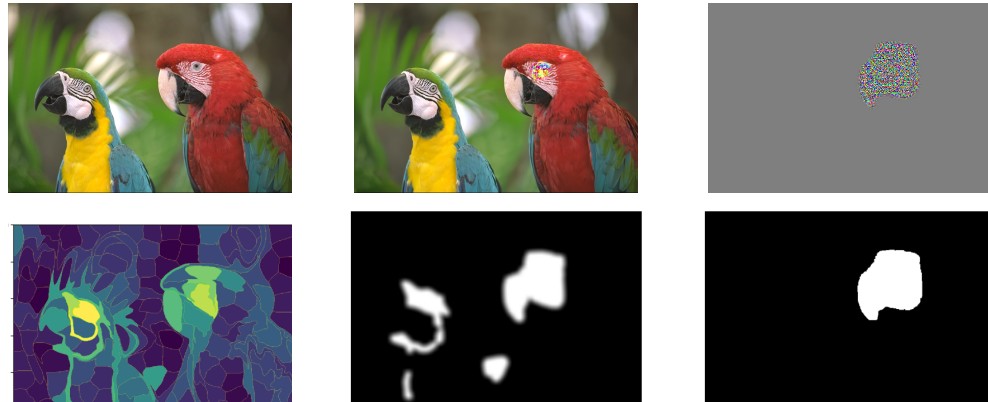

Figure 6: kodim23, Cheng2020-anchor, Maxdistortion attack. Top: the attacked image, model's output (decompressed image), attack noise pattern. Bottom: entropy regions, smoothed mask, final mask. **PSNR(oi, oo) = 39.14 dB**, **PSNR(ai, ao) = 28.88 dB** , **BPP(oc) = 0.43**, **BPP(ac) = 0.48**, where oi, oo, oc - original (input, output, compressed) image, and ai, ao, ac - attacked (input, output, compressed) image.

## D ATTACKS AND DEFENSE ON CHENG2020-ATTN MODEL

In addition to the two original victim models, we have extended our experiments to include the Cheng attention model at quality level 6. Table 9 below illustrates the results, showing that our attack performs consistently well, achieving a significant degradation in quality metrics for the attacked images while maintaining imperceptible noise. This further demonstrates the robustness and generalizability of our method across different AI compression models.

Using our attack method, we observed that the average BPP(ac) closely matches the BPP(oc) (oc, ac - original and attacked images' compressions), demonstrating the efficiency of the attack in preserving file size. Despite an average PSNR drop of 11.63, the high PSNR(oi, ai) of 44.66 ensures that the attacked images remain visually similar to the original ones. Furthermore, as detailed in the defense section, we successfully countered the attack, achieving a PSNR(di, do) of 36.13, which is comparable to the baseline (di, do - defended input and output images). This was accomplished while maintaining a high PSNR between the defended images and the original ones, underscoring the robustness of our defense approach.

## E FIGURES FOR ATTACKS TO TCM MODEL

Figure 7 represents TCM model (N128) Log-Exp attack. Image: kodim14. Left to right: optimized noise, attacked image, TCM output. More attacked images can be found in the collage 8

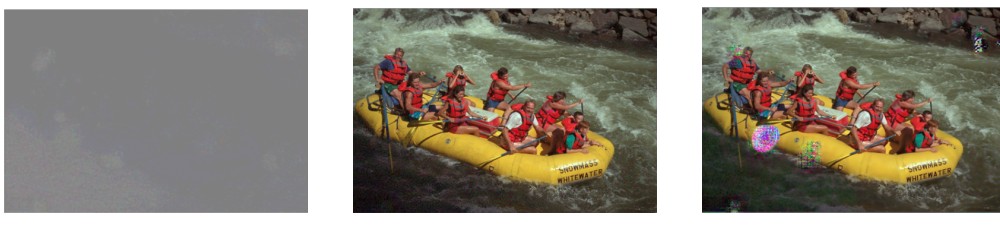

Figure 7: TCM model (N128) Log-Exp attack. Image: kodim14. Left to right: optimized noise, attacked image, TCM output.

Table 9: PSNR and BPP metrics for all images from Kodak dataset using Cheng2020-attn model with quality 6. Abbreviations: oi, oo, oc - original (input, output, compressed) image; ai, ao, ac - attacked (input, output, compressed) image; di, do, dc - defended (input, output, compressed) image.

| Image | PSNR(oi, oo) | PSNR(ai, ao) | PSNR(oi, ai) | BPP(oc) | BPP(ac) | PSNR(di, do) | PSNR(di, oi) | BPP(dc) |
|---|---|---|---|---|---|---|---|---|
| kodim01 | 35.08 | 29.92 | 34.67 | 1.41 | 1.42 | 34.67 | 45.52 | 1.42 |
| kodim02 | 36.85 | 18.83 | 41.31 | 0.70 | 0.72 | 35.95 | 42.50 | 0.71 |
| kodim03 | 39.35 | 22.78 | 46.84 | 0.50 | 0.52 | 38.82 | 47.49 | 0.52 |
| kodim04 | 37.25 | 23.97 | 48.16 | 0.70 | 0.71 | 37.03 | 49.43 | 0.70 |
| kodim05 | 35.70 | 23.89 | 43.04 | 1.35 | 1.38 | 34.96 | 43.35 | 1.38 |
| kodim06 | 36.48 | 30.24 | 46.58 | 1.07 | 1.07 | 35.68 | 47.22 | 1.08 |
| kodim07 | 39.15 | 25.13 | 44.10 | 0.61 | 0.64 | 38.24 | 44.33 | 0.64 |
| kodim08 | 34.61 | 28.41 | 43.47 | 1.45 | 1.46 | 34.13 | 43.98 | 1.46 |
| kodim09 | 38.70 | 25.09 | 53.32 | 0.52 | 0.52 | 38.58 | 54.16 | 0.52 |
| kodim10 | 38.52 | 24.12 | 46.66 | 0.56 | 0.57 | 38.11 | 47.75 | 0.57 |
| kodim11 | 36.50 | 24.76 | 42.48 | 0.93 | 0.94 | 34.82 | 43.03 | 0.94 |
| kodim12 | 38.48 | 20.89 | 50.06 | 0.57 | 0.57 | 38.46 | 57.12 | 0.57 |
| kodim13 | 32.49 | 25.65 | 44.27 | 1.80 | 1.81 | 31.79 | 44.60 | 1.81 |
| kodim14 | 35.49 | 25.27 | 41.90 | 1.17 | 1.19 | 34.89 | 42.24 | 1.19 |
| kodim15 | 37.35 | 22.05 | 48.10 | 0.65 | 0.67 | 37.13 | 49.17 | 0.67 |
| kodim16 | 37.96 | 24.29 | 44.25 | 0.75 | 0.77 | 37.25 | 44.62 | 0.77 |
| kodim17 | 37.70 | 22.98 | 47.05 | 0.65 | 0.66 | 37.20 | 48.13 | 0.65 |
| kodim18 | 34.49 | 26.18 | 42.98 | 1.15 | 1.16 | 32.62 | 63.65 | 1.16 |
| kodim19 | 36.77 | 25.14 | 44.12 | 0.86 | 0.88 | 36.05 | 44.47 | 0.86 |
| kodim20 | 38.27 | 22.27 | 47.11 | 0.59 | 0.61 | 37.95 | 48.28 | 0.60 |
| kodim21 | 36.45 | 30.45 | 44.31 | 0.91 | 0.92 | 35.84 | 44.76 | 0.92 |
| kodim22 | 36.01 | 25.70 | 43.81 | 0.94 | 0.95 | 35.61 | 44.24 | 0.95 |
| kodim23 | 39.14 | 28.88 | 40.96 | 0.43 | 0.48 | 37.02 | 41.22 | 0.47 |
| kodim24 | 35.10 | 27.93 | 42.23 | 1.13 | 1.15 | 34.20 | 42.49 | 1.15 |
| Average | 36.83 | 25.20 | 44.66 | 0.89 | 0.91 | 36.13 | 46.82 | 0.90 |

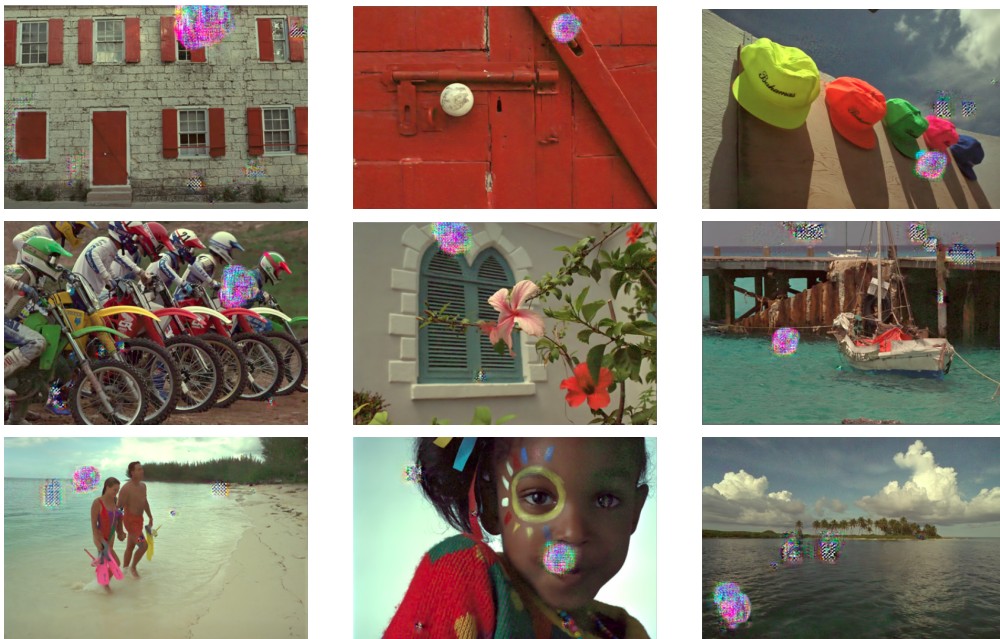

Figure 8: TCM model (N128) Log-Exp noise attack compression-decompression outputs collage.

## F MAX(BPP) ATTACK AND DEFENSE

In addition, we tested another type of attack using the same algorithm, called MaxBPP attack, The goal is to increase the file size without introducing artifacts in the decompressed image, ensuring that PSNR(ai, ao) matches PSNR(oi, oo). Attack and defense for kodim07 image are presented in the figure 9. The results using Cheng-anchor model with quality 6 are illustrated in table 10.

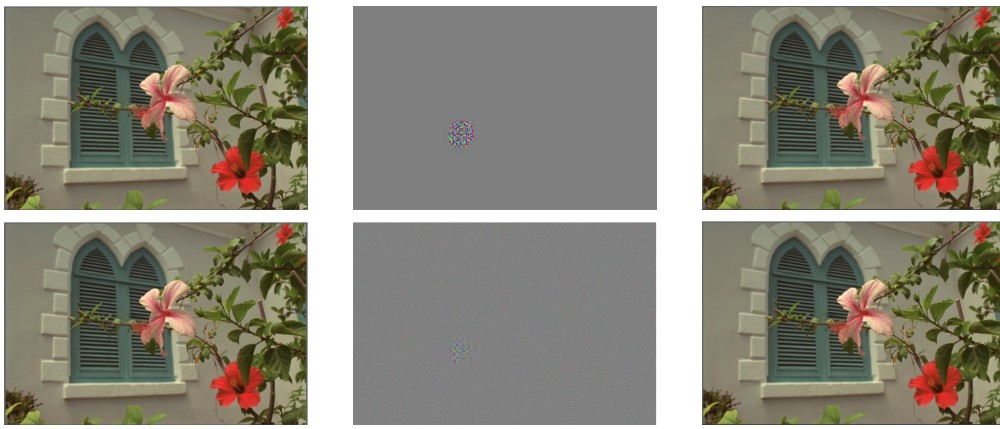

Figure 9: Image: kodim07, model: cheng2020-anchor quality 6. Top: Max(BPP) attacked image, attack noise, model output. Bottom: defended image, defnsive noise, model output of the defended image.

Table 10: MaxBPP attack and defense using Cheng-anchor model with quality 6 for all images in Kodak dataset. Abbreviations: oi, oo, oc - original (input, output, compressed) image; ai, ao, ac - attacked (input, output, compressed) image.

| Kodim | BPP(oc) | BPP(ac) | PSNR(oi, ai) | PSNR(oi, oo) | PSNR(ai, ao) | PSNR(di, do) | PSNR(di, oi) | BPP(dc) |
|---|---|---|---|---|---|---|---|---|
| kodim01 | 1.42 | 8.93 | 46.42 | 35.22 | 35.18 | 34.86 | 44.00 | 1.44 |
| kodim02 | 0.69 | 7.83 | 44.17 | 36.93 | 36.45 | 36.03 | 42.41 | 0.66 |
| kodim03 | 0.51 | 7.15 | 39.82 | 39.53 | 39.49 | 37.33 | 40.95 | 0.51 |
| kodim04 | 0.70 | 7.30 | 40.93 | 37.31 | 40.93 | 35.34 | 38.48 | 0.72 |
| kodim05 | 1.33 | 7.72 | 42.72 | 35.77 | 35.45 | 35.15 | 41.63 | 1.36 |
| kodim06 | 1.06 | 5.32 | 36.60 | 36.60 | 42.51 | 36.43 | 41.34 | 1.05 |
| kodim07 | 0.60 | 7.05 | 49.19 | 39.30 | 39.13 | 38.87 | 48.56 | 0.61 |
| kodim08 | 1.44 | 6.67 | 48.56 | 34.83 | 34.58 | 34.75 | 49.24 | 1.45 |
| kodim09 | 0.52 | 6.85 | 43.10 | 38.72 | 38.49 | 38.46 | 41.83 | 0.57 |
| kodim10 | 0.56 | 6.38 | 46.05 | 38.58 | 38.35 | 37.68 | 43.45 | 0.59 |
| kodim11 | 0.93 | 7.32 | 41.14 | 36.60 | 35.99 | 35.61 | 40.14 | 1.06 |
| kodim12 | 0.57 | 4.48 | 51.35 | 38.50 | 38.44 | 37.74 | 44.84 | 0.57 |
| kodim13 | 1.18 | 4.67 | 40.32 | 32.39 | 32.37 | 31.67 | 43.61 | 1.82 |
| kodim14 | 1.17 | 6.50 | 49.16 | 35.60 | 35.57 | 35.47 | 48.87 | 1.18 |
| kodim15 | 0.65 | 7.90 | 48.32 | 37.38 | 37.26 | 36.84 | 44.54 | 0.65 |
| kodim16 | 0.75 | 6.74 | 52.01 | 38.01 | 37.80 | 38.00 | 50.04 | 0.75 |
| kodim17 | 0.64 | 5.60 | 50.86 | 37.74 | 37.65 | 37.12 | 44.96 | 0.64 |
| kodim18 | 1.15 | 7.74 | 46.75 | 43.43 | 34.41 | 34.16 | 43.51 | 1.22 |
| kodim19 | 0.85 | 7.07 | 46.02 | 36.87 | 36.62 | 36.17 | 43.30 | 0.88 |
| kodim20 | 0.59 | 5.03 | 42.84 | 38.33 | 37.66 | 36.19 | 39.33 | 0.66 |
| kodim21 | 0.92 | 4.84 | 42.59 | 36.55 | 36.29 | 35.87 | 41/01 | 0.97 |
| kodim22 | 0.94 | 9.95 | 44.73 | 36.10 | 35.83 | 35.50 | 42.97 | 0.98 |
| kodim23 | 0.43 | 5.49 | 43.93 | 39.28 | 38.68 | 37.93 | 41.85 | 0.47 |
| kodim24 | 1.13 | 6.35 | 41.15 | 35.11 | 34.90 | 33.27 | 36.23 | 1.23 |
| Avg(ours) | 0.86 | 6.70 (8.62x) | 44.95 | 37.28 | 37.09(-0.5%) | 36.10 | 43.21 | 0.92 |
| Avg(Liu et.alLiu et al. (2022)) | 0.86 | (19.83x) | - | 37.28 | 21.25(-43%) | - | - | - |
| FactorAtt(Liu et.alLiu et al. (2022)) | 0.90 | 2.38 (2.64x) | - | 35.05 | - | - | - | - |

Table 11 presents the impact of applying a MaxBPP attack and defense using Cheng-anchor quality 3. The results indicate a significant increase in BPP(ac) values compared to the original, achieving an effective manipulation of file size. Despite this increase, the PSNR(oi, ai) values remain close to acceptable levels, which ensures that the perceptual quality is mostly preserved.

In comparison to Liu et al. (2022), our results using Cheng-anchor with quality 6 show an average 8.62x increase in BPP for the attacked images compared to the original ones, while maintaining a very high PSNR(oi, ai) that ensures the similarity among them. In contrast, Liu et al. (2022) achieved a 19.83x increase in BPP but at the cost of a -43% drop in PSNR(ai, ao), rendering the attacked images unrealistic and noticeably different from the originals with clear artifacts in the decompressed attacked images. Figure 9 shows an example of MaxBPP attack and defense. Rather than developing a defense algorithm,Liu et al. (2022) trained a new model, FactorAttn, and evaluated its performance against the same attack they proposed. From the table (last row), it is evident that their approach resulted in only a 2.64x increase in BPP(ac) compared to BPP(oc). However, they did not compute or report any metrics to demonstrate the similarity between the attacked image and the original image.

For quality 3, Liu et al. (2022) achieved a 9.9x increase in BPP but with a -5.1% reduction in PSNR(ai, ao). In comparison, our method achieved a 5.82x increase in BPP with only -3.3% reduction in PSNR(oi, ai). while preserving a high PSNR(oi, ai), indicating that the attacked images remain visually indistinguishable from the originals, with the applied noise being imperceptible to the human eye.

Instead of designing a defense algorithm, Liu et al. (2022) also trained a new model, FactorAttn, and assessed its performance against their proposed attack. As shown in the table (last row), their method achieved only a 1.96x increase in BPP(ac) compared to BPP(oc). However, they did not provide any metrics to evaluate the similarity between the attacked image and the original image.

Table 11: MaxBPP Attack and defense using Cheng-anchor model with quality 3 for all images in Kodak dataset

| Index | BPP(oc) | BPP(ac) | PSNR(oi, ai) | PSNR(oi, oo) | PSNR(ai, ao) | PSNR(di, do) | PSNR(di, oi) | BPP(dc) |
|---|---|---|---|---|---|---|---|---|
| kodim01 | 0.53 | 1.80 | 33.31 | 29.18 | 28.54 | 25.60 | 27.15 | 0.65 |
| kodim02 | 0.14 | 2.19 | 36.11 | 32.31 | 31.51 | 28.98 | 30.87 | 0.29 |
| kodim03 | 0.20 | 2.85 | 35.29 | 34.56 | 32.75 | 29.60 | 30.62 | 0.24 |
| kodim04 | 0.25 | 1.31 | 32.17 | 32.59 | 31.14 | 27.73 | 28.27 | 0.31 |
| kodim05 | 0.57 | 1.89 | 39.82 | 29.81 | 29.71 | 29.08 | 35.77 | 0.58 |
| kodim06 | 0.40 | 0.93 | 33.87 | 30.53 | 29.72 | 29.17 | 34.38 | 0.41 |
| kodim07 | 0.27 | 1.59 | 29.22 | 34.18 | 33.43 | 33.24 | 32.15 | 0.28 |
| kodim08 | 0.59 | 2.63 | 39.29 | 29.12 | 28.94 | 28.01 | 33.71 | 0.60 |
| kodim09 | 0.23 | 1.17 | 34.21 | 34.34 | 32.35 | 32.86 | 33.70 | 0.26 |
| kodim10 | 0.24 | 0.96 | 33.35 | 34.14 | 33.54 | 32.93 | 31.54 | 0.25 |
| kodim11 | 0.33 | 1.86 | 31.78 | 31.16 | 30.23 | 29.80 | 32.09 | 0.35 |
| kodim12 | 0.20 | 0.90 | 33.76 | 33.79 | 32.30 | 32.99 | 33.36 | 0.22 |
| kodim13 | 0.69 | 3.22 | 36.25 | 26.70 | 26.56 | 26.35 | 35.47 | 0.71 |
| kodim14 | 0.43 | 2.36 | 31.17 | 30.19 | 28.90 | 29.39 | 30.75 | 0.45 |
| kodim15 | 0.24 | 4.97 | 35.84 | 32.69 | 31.62 | 30.11 | 30.94 | 0.25 |
| kodim16 | 0.27 | 2.48 | 35.52 | 32.45 | 31.70 | 30.04 | 32.22 | 0.28 |
| kodim17 | 0.26 | 1.23 | 31.23 | 32.99 | 32.40 | 30.24 | 30.28 | 0.29 |
| kodim18 | 0.45 | 1.27 | 32.64 | 29.54 | 28.67 | 28.96 | 29.88 | 0.48 |
| kodim19 | 0.30 | 1.41 | 30.02 | 31.67 | 29.87 | 29.99 | 31.98 | 0.31 |
| kodim20 | 0.22 | 1.70 | 36.49 | 33.44 | 32.76 | 32.67 | 30.95 | 0.23 |
| kodim21 | 0.37 | 1.36 | 30.72 | 31.14 | 29.66 | 29.48 | 34.13 | 0.38 |
| kodim22 | 0.33 | 1.86 | 33.81 | 31.02 | 30.24 | 30.47 | 37.35 | 0.34 |
| kodim23 | 0.19 | 0.99 | 32.90 | 35.20 | 32.22 | 33.51 | 37.36 | 0.20 |
| kodim24 | 0.45 | 4.50 | 31.64 | 29.47 | 28.33 | 30.06 | 29.76 | 0.57 |
| Avg(ours) | 0.34 | 1.98 (5.82x) | 33.77 | 31.76 | 30.71(-3.3%) | 30.05 | 32.28 | 0.37 |
| Avg(Liu et.alLiu et al. (2022)) | 0.34 | (9.9x) | - | 31.76 | 30.14(-5.1%) | - | - | - |
| FactorAtt(Liu et.alLiu et al. (2022)) | 0.26 | 0.51 (1.96x) | - | 29.59 | - | - | - | - |

## G  ATTACKS FOR IMAGE CLASSIFICATION

We extended our analysis to a classification task as a complementary experiment. Using EfficientNet and a class from the COCO dataset with the highest confidence prediction ("tubby cat," class 281), we applied our attack. Post-attack, the model misclassified the image as class 933. The applied noise remained imperceptible, with a high PSNR of 32.77 dB without using the mask, while it was 41.59 using the entropy mask and the attack reduced the classification confidence to 9.03% (lower is better).

For comparison, applying the FGSM attack on the same task yielded a PSNR of 13.77 dB with a significantly higher confidence of 48.77%, underscoring the efficiency and subtlety of our method. Figure 10

## H  EVALUATION ON EXTERNAL IMAGES

We extended our experiments to include images from the COCO dataset. This allowed us to evaluate the generalizability of our approach on a broader set of natural images. Figure 11 illustrates the results for a natural image (giraffe) from the COCO dataset, including the defense algorithm's output. Table 12 summarizes the metrics for this experiment. Key observations include:

- A significant drop in PSNR ($-8$ dB) for the attacked image while maintaining the same bits per pixel (bpp) rate.

- The PSNR between the original and attacked image remains high, demonstrating that the applied noise is imperceptible to the human eye.

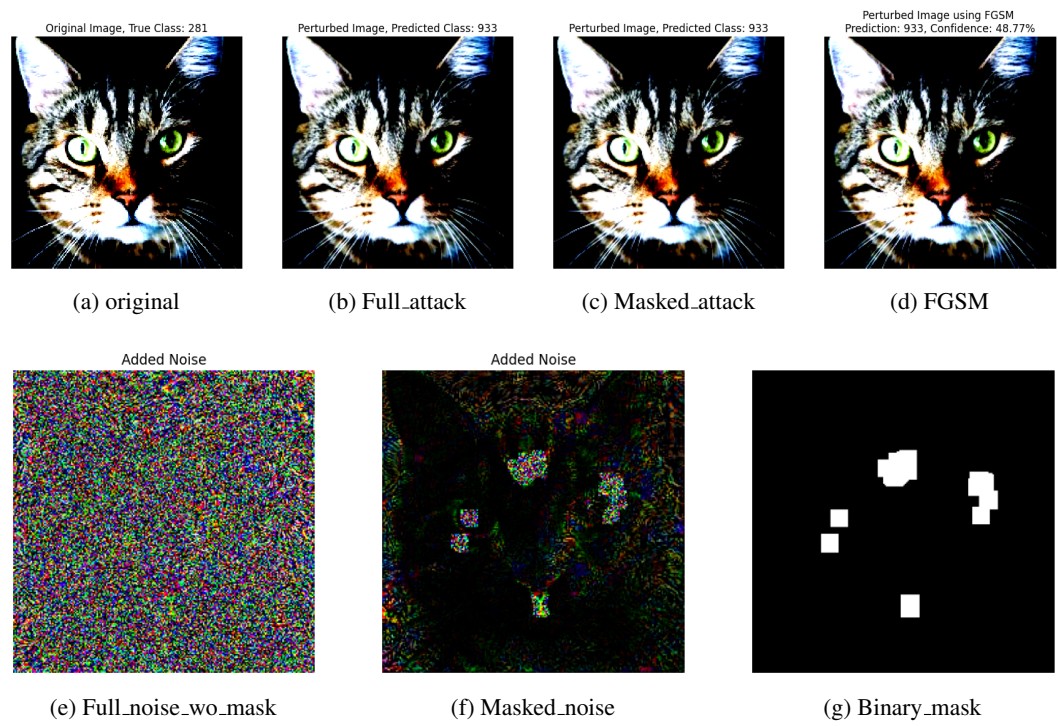

<table>
<tr><td>(a) original</td><td>(b) Full_attack</td><td>(c) Masked_attack</td><td>(d) FGSM</td></tr>
</table>

(e) Full_noise_wo_mask      (f) Masked_noise      (g) Binary_mask

Figure 10: Local vs spatial vs FGSM attacks for an image form Coco dataset

- The defense algorithm successfully restores the image quality, achieving results close to the original image with minimal artifacts.

Table 12: Metrics for the attack of the Giraffe image from the COCO dataset, Cheng2020-anchor model, quality 6. Abbreviations: oi - original input; ai, ac, ao - attacked (input, compressed, output) image.

| Method | PSNR(ai, ao) | PSNR(oi, ai) | BPP(ac) | SSIM(ao, oi) | VIF(oi, ai) | VIF(oi, ao) |
|---|---|---|---|---|---|---|
| baseline_full | 26.77 | | 2.36 | 0.97 | 1.00 | 0.98 |
| minpsnr_ | 18.17 | 42.74 | 2.36 | 0.95 | 0.99 | 0.75 |
| highentropy_minpsnr_masksmooth | 17.53 | 42.87 | 2.36 | 0.94 | 0.99 | 0.73 |
| highentropy_minpsnr_maskshrink | 19.08 | 48.17 | 2.36 | 0.95 | 0.99 | 0.78 |
| def_minpsnr | 26.78 | 48.82 | 2.36 | 0.97 | 1.00 | 0.98 |

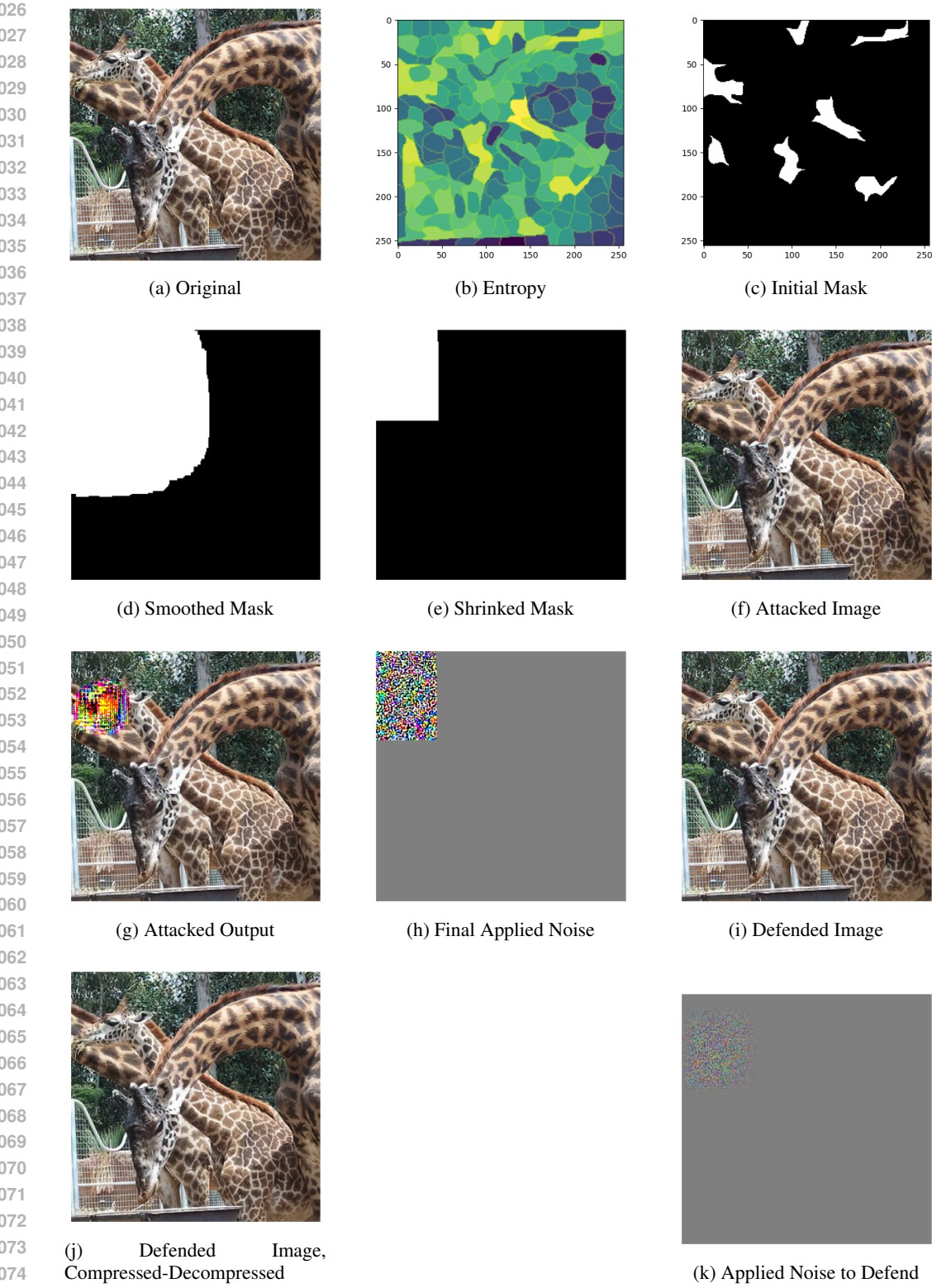

(a) Original

(b) Entropy

(c) Initial Mask

(d) Smoothed Mask

(e) Shrinked Mask

(f) Attacked Image

(g) Attacked Output

(h) Final Applied Noise

(i) Defended Image

(j)     Defended     Image,
Compressed-Decompressed

(k) Applied Noise to Defend

Figure 11: Results of various stages in the attack and defense process for the Giraffe example, Cheng2020-anchor model, quality 6.

