# OpenReview forum: "Adaptive Log-Exp Perturbations for Secure AI Image Compression"
_ICLR.cc/2025/Conference — ICLR 2025 Conference Withdrawn Submission_

### Official Review · Reviewer_FnDw · 2024-11-02

**Soundness:** 1
**Presentation:** 2
**Contribution:** 1
**Rating:** 1
**Confidence:** 5

**Summary:**

AI-driven image compression surpasses traditional methods in efficiency and quality but is still susceptible to adversarial attacks. Most attacks involve adding uniform, small perturbations across image pixels, yet these can be more noticeable in bright regions where visual sensitivity is higher. This observation inspires a novel approach that minimizes visible noise using adaptive perturbations.
The proposed nonlinear log-exp perturbation strategy applies more noise to dark pixels while sparing brighter areas. Evaluations show this method can distort decompression model outputs and increase bit rates without affecting perceptual quality, offering new protections for AI-based image compression systems.

**Strengths:**

1. This study explores a new perturbation space for adversarial attacks targeting learned image compression.

2. The paper proposes learning an optimal noise pattern to "correct" adversarial examples back to clean images as a defense against such attacks.

**Weaknesses:**

1. The novelty of the proposed adversarial attacks is limited, and the scenario is restricted to deep image compression specifically.

2. The defense strategy appears overly simplistic. In a white-box scenario, this defense would likely fail, as attackers would know the protective noise patterns and could subtract it to carry out a successful attack. Even in a black-box scenario, attackers could use multiple queries to optimize their attack against the protective noise pattern. Overall, this approach does not fundamentally enhance the model's inherent adversarial robustness.

3. The experiments are not sufficient, since only two victim models are included.

**Questions:**

See weakness above.

---

### Official Review · Reviewer_Wd5Q · 2024-11-02

**Soundness:** 1
**Presentation:** 1
**Contribution:** 2
**Rating:** 3
**Confidence:** 4

**Summary:**

This paper proposes a new adversarial attack for neural image compression models. This method is inspired by Weber's law and applies more noise to pixels in dark regions, where the human eye is less sensitive to variations, and less noise to brighter regions. Additionally, this paper proposes a defense strategy to reduce the effect of adversarial noise on the performance of neural image compression models. The authors analyze their attacks and defenses on the Cheng2020 anchor model and the TCM model using the Kodak dataset.

**Strengths:**

1. It's a good idea to design adversarial perturbations to exploit human perception both in terms of luminance and high-entropy regions.

**Weaknesses:**

1. The writing and presentation make this paper hard to understand. For example, Tables 2 and 3 show results for 24 individual images without aggregating, the figure captions are not clear, the equations are not numbered, and the notation is not consistent throughout.
2. The method sections for both the adversarial attacks and defense strategies define an objective to optimize, but do not explain the methods used to solve these. I'm not sure how the models are trained or how hyper-parameters (e.g., $\delta, \lambda, \kappa$) are chosen.
3. Results are only shown on Kodak dataset (24 images).
4. There is no related works section. Paper does not address closely related work (e.g., Liu et al., MALICE: Manipulation Attacks on Learned Image ComprEssion, 2022; Lei et al., Out-of-Distribution Robustness in Deep Learning Compression, 2021; Lieberman et al., Neural Image Compression: Generalization, Robustness, and Spectral Biases, 2023).
5. There are contradictions about important concepts in the paper. For example, line 15 says "the human eye is less sensitive to variations in dark areas than in bright ones, making noise in brighter areas more visible" while line 140 says, "the human visual system is more sensitive to brightness changes in dark regions compared to brighter regions, where the eye struggles to discern small variations." Similarly, lines 18, 107, 131, 175 all imply that humans are less sensitive to light changes in dark regions while line 178 states that humans are more sensitive to changes in dark regions.

**Questions:**

1. On lines 126-140, the authors discuss Weber's law. They state that the ratio $\frac{\delta I}{I}$ (i.e., the ratio between the Just Noticeable Difference (JND) and the original luminance) decreases as the luminance increases. My understanding of Weber's law is that the ratio $\frac{\delta I}{I}$ is constant, but the JND increases as the luminance increases [1]. Could the authors please clarify this?
2. Where does the 2.3 on line 148 come from? There is no citation for the Weber-Fechner Law.
3. Why is $\mathbf{n}$ used for both the adversarial noise (line 217) and optimal noise pattern (line 266)?
4. Why is the attack objective to minimize $PSNR(f(\mathbf{x}^*), \mathbf{x})$ instead of $PSNR(f(\mathbf{x}^*), \mathbf{x^*})$ ? Doesn't this require that the model know what the perturbation is (i.e., the difference between $\mathbf{x}$ and $\mathbf{x^*}$)?

[1] https://www.britannica.com/science/Webers-law

---

### Official Review · Reviewer_cCjX · 2024-11-04

**Soundness:** 2
**Presentation:** 1
**Contribution:** 2
**Rating:** 3
**Confidence:** 4

**Summary:**

This paper introduces a novel approach to adversarial attacks in AI-based image compression systems, leveraging adaptive, nonlinear log-exp perturbations. The technique exploits human visual perception by minimizing noise in brighter areas and concentrating perturbations in darker regions. The authors present both an attack strategy and a corresponding defense mechanism to mitigate the attack’s impact. Experiments demonstrate the approach's efficacy in degrading image quality while remaining undetectable by the human eye, as well as the effectiveness of the defense in preserving image quality.

**Strengths:**

- The proposed log-exp perturbation model effectively leverages human visual perception, an innovative approach that enhances the imperceptibility of adversarial attacks.

- The paper proposes a simple defense solution against adversarial attacks, without the need for additional training constraints on the model.

- The experiments support the effectiveness of both the attack and the defense methods.

**Weaknesses:**

- At a high level, the paper's experiments and structure appear quite disorganized. For instance, the primary focus of the paper is unclear, and the connection between the proposed attack and defense methods lacks clarity. The defense is not customized and adapted for the proposed attack, and could potentially be applied to other types of adversarial attacks. However, the paper only tests it against its own proposed attack, leaving the evaluation of the defense significantly incomplete.

- The experimental results in tables and figures are presented on a per-sample basis rather than averaged across a larger dataset, making it difficult to draw generalized conclusions.

- The paper does not demonstrate the proposed attack's superiority over existing adversarial attacks. Providing visual (and numerical) comparisons would allow readers to verify the attack's higher imperceptibility to the human eye.

- From the experiments shown, it appears that the proposed attack can be fully neutralized by the defense, raising questions about the purpose of creating an attack more imperceptible to the human eye, if it is easily defended. If this is not the case (for instance, if a stronger attack perturbation could still succeed), a comprehensive evaluation that includes current state-of-the-art attacks and defenses would be essential to establish the superiority of the proposed attack and defense.

**Questions:**

- Please refer to the weaknesses.

---

### Official Review · Reviewer_RZoN · 2024-11-08

**Soundness:** 3
**Presentation:** 2
**Contribution:** 2
**Rating:** 5
**Confidence:** 4

**Summary:**

The paper proposes a perceptually inspired approach for attacks on deep image compression, adding noise to the illuminance channel of images to reduces the quality of reconstructed images or to increase bitrate.

**Strengths:**

Attacks on image compression are an emerging field and interesting topics for further exploration.

**Weaknesses:**

It's unclear to me why the attack is specifically targeted only towards deep image compression and not more generally targeted towards other image tasks as well (classification, segmentation etc.) There is obviously a very large body of work in these domains, and nothing in the proposed idea of adding illuminance noise is specific to compression; evaluating the attack more generally will also allow for a more thorough comparison with the large body of work on adversarially imperceptible attacks. Second, "Manipulation Attacks on Learned Image Compression" from IEEE Trans. on AI addresses the same problem and should be compared against.

**Questions:**

- Please implement your attacks for image classification tasks as well, and compare against the best known attacks in those domains.
- Please compare your results against the IEEE TAI paper.

---

### Note · Authors · 2025-01-11

I have read and agree with the venue's withdrawal policy on behalf of myself and my co-authors.